# Exploring Renal Changes after Bariatric Surgery in Patients with Severe Obesity

**DOI:** 10.3390/jcm11030728

**Published:** 2022-01-29

**Authors:** Anna Oliveras, Susana Vázquez, María José Soler, Isabel Galceran, Xavier Duran, Albert Goday, David Benaiges, Marta Crespo, Julio Pascual, Marta Riera

**Affiliations:** 1Nephrology Department, Hospital Universitari del Mar, 08003 Barcelona, Spain; svazquez@hospitaldelmar.cat (S.V.); igalceran@hospitaldelmar.cat (I.G.); mcrespo@hospitaldelmar.cat (M.C.); julpascual@gmail.com (J.P.); 2IMIM, Hospital del Mar Medical Research Institute, 08003 Barcelona, Spain; xduran@imim.es (X.D.); agoday@hospitaldelmar.cat (A.G.); dbenaiges@hospitaldelmar.cat (D.B.); mriera1@imim.es (M.R.); 3Department of Experimental and Health Sciences, Area of Medicine, Universitat Pompeu Fabra, 08002 Barcelona, Spain; 4Red de Investigación Renal (REDINREN), Instituto Carlos III-FEDER, 28029 Madrid, Spain; 5Nephrology Research Group, Vall d’Hebron Research Institute (VHIR), Nephrology Department, Hospital Universitari Vall d’Hebron, Universitat Autònoma de Barcelona, 08035 Barcelona, Spain; mjsoler01@gmail.com; 6Endocrinology Department, Hospital Universitari del Mar, 08003 Barcelona, Spain; 7Medicine Department, Universitat Autònoma de Barcelona, 08193 Barcelona, Spain

**Keywords:** bariatric surgery, hyperfiltration, albuminuria, renin-angiotensin axis, aldosterone, glucose metabolism

## Abstract

Obesity-related hyperfiltration leads to an increased glomerular filtration rate (GFR) and hyperalbuminuria. These changes are reversible after bariatric surgery (BS). We aimed to explore obesity-related renal changes post-BS and to seek potential mechanisms. Sixty-two individuals with severe obesity were prospectively examined before and 3, 6 and 12 months post-BS. Anthropometric and laboratory data, 24 h-blood pressure, renin-angiotensin-aldosterone system (RAS) components, adipokines and inflammatory markers were determined. Both estimated GFR (eGFR) and albuminuria decreased from the baseline at all follow-up times (*p*-for-trend <0.001 for both). There was a median (IQR) of 30.5% (26.2–34.4) reduction in body weight. Plasma glucose, glycosylated hemoglobin, fasting insulin and HOMA-index decreased at 3, 6 and 12 months of follow-up (*p*-for-trend <0.001 for all). The plasma aldosterone concentration (median (IQR)) also decreased at 12 months (from 87.8 ng/dL (56.8; 154) to 65.4 (56.8; 84.6), *p* = 0.003). Both leptin and hs-CRP decreased (*p* < 0.001) and adiponectine levels increased at 12 months post-BS (*p* = 0.017). Linear mixed-models showed that body weight (coef. 0.62, 95% CI: 0.32 to 0.93, *p* < 0.001) and plasma aldosterone (coef. −0.07, 95% CI: −0.13 to −0.02, *p* = 0.005) were the independent variables for changes in eGFR. Conversely, glycosylated hemoglobin was the only independent variable for changes in albuminuria (coef. 0.24, 95% CI: 0.06 to 0.42, *p* = 0.009). In conclusion, body weight and aldosterone are the main factors that mediate eGFR changes in obesity and BS, while albuminuria is associated with glucose homeostasis.

## 1. Introduction

Worldwide obesity has nearly tripled since 1975 [1]. Obesity is associated with an increase in morbidity and mortality, mainly from cardiovascular disease and diabetes, among others [2]. In addition, obesity is also a major risk factor for the development of chronic kidney disease [3]. Obesity is usually characterized by an abnormally high glomerular filtration rate (GFR) and long-lasting hyperfiltration may cause renal lesions similar to secondary focal segmental glomerulosclerosis, the so-called obesity-related glomerulopathy, leading to a decrease in GFR and hyperalbuminuria [4]. In patients with severe obesity, both hyperfiltration and albuminuria are a consequence of increased intraglomerular pressure and glomerular surface area. Although the prevalence of microalbuminuria or proteinuria is higher in obese patients who have diabetes, albuminuria is still higher in non-diabetic obese patients than in the general population [5]. Finally, increased body mass index (BMI) has been linked to a loss of renal function, as well as a higher risk of end-stage renal disease [5]. In severely obese patients, lifestyle changes often fail to significantly reduce body weight. Fortunately, bariatric surgery (BS), a surgical approach that can be performed through different technical procedures, promotes weight loss so that it achieves much better results. Beyond weight loss, BS leads to the improvement of various obesity-related diseases such as hypertension, disorders of glucose and lipid metabolism, obstructive sleep apnea or non-alcoholic steatohepatitis, among others. More importantly, a recent meta-analysis [6] showed a reduction in major adverse cardiovascular events in patients with obesity and cardiovascular disease who underwent BS compared with those who did not have surgery. Recently, obesity-induced hyperfiltration and albuminuria have been shown to be reversible after bariatric surgery [4,7]. Altered renal haemodynamics as well as a deleterious adipocytokine pattern favored by obesity appear to be at the root of both obesity-related renal impairment and its improvement after BS, at least in diabetics [4,8,9].

Here we sought to assess changes in renal function at different follow-up times in patients with severe obesity undergoing BS. In addition, we explored the possible role of the renin-angiotensin aldosterone system (RAS), changes in glucose metabolism, and inflammation as potential mechanisms mediating these changes.

## 2. Materials and Methods

### 2.1. Methods

#### 2.1.1. Study Design and Patients

The BARIHTA study is a prospective observational trial in a cohort of consecutively enrolled patients with severe obesity scheduled to undergo BS (clinicaltrials.gov identifier: NCT03115502). Details about BARIHTA trials have been previously published [10]. Thus, the BARIHTA study prospectively recruited outpatients with severe obesity who went to consultations at the Hospital del Mar (Barcelona, Catalonia, Spain) seeking surgical treatment. All individuals (both sexes, aged 18–60 years) with a medical indication for surgical intervention and who agreed to undergo the treatment with BS were invited to participate. Indications for BS included those patients with a body mass index (BMI) > 40 kg/m^2^ or grade II obesity (BMI > 35 kg/m^2^) plus associated comorbidities (i.e., type 2 diabetes mellitus, obesity-associated hypoventilation disorders, high blood pressure, or dyslipidemia). Patients with any endocrine disease causing obesity or severe psychiatric diseases were excluded. Detailed information on the trial was provided by qualified professionals of the Hypertension and Vascular Risk Unit (Nephrology Department, Hospital del Mar). The exclusion criteria comprised the ruling out of the BS program for any reason or the refusal to give consent. The trial was approved by the local institutional Ethic Committee in accordance with the Declaration of Helsinki, and written informed consent was obtained from all participants.

Here we evaluate and report the effects of BS on renal function and explore its possible mechanisms by analyzing its relationship with various components of the renin-angiotensin-aldosterone system (RAS), along with inflammatory markers and adipokines, as previously specified per protocol.

Demographic, anthropometric and clinical data were recorded from all participants as a baseline. Anthropometric characteristics, pharmacological treatment, office- and 24 h-ambulatory-blood pressure (BP) recordings, routine laboratory tests, including renal function as assessed by the estimated glomerular filtration rate (eGFR) and by determining albuminuria, and determinations of components of the RAS, adipokines and inflammatory markers, were obtained at baseline and 1, 3, 6 and 12 months after surgery. Changes at follow-ups are evaluated from three months on to avoid the major hemodynamic instability one month after BS. Hypertension was considered if previously diagnosed and/or if the baseline 24 h-BP was ≥130/80 mmHg. Diabetes mellitus (DM) was considered if the patient received antidiabetic treatment or had ≥2 fasting plasma glucose determinations ≥126 mg/dL or if glycosylated haemoglobin A1c was >6.5%.

About one-third of the study population was under treatment with at least one drug that interfered with the RAS. Given that these drugs could introduce a bias by interfering with renal function parameters, the main analyzes were performed separately in both the entire cohort and in the untreated patients.

#### 2.1.2. Procedures

##### Blood Pressure Measurements

Brachial-BP measurements and calculation of central-BP and other arterial parameters through the oscillometric method (ARCSolver algorithm) were obtained from a Mobil-O-Graph^®^ NG-ambulatory blood pressure (NG-ABPM) device by IEM, Stolberg, Germany. The monitor was placed on a working day between 08:00–10:00 h A.M., and after a 5 min rest, BP was consecutively determined four times at 1-min intervals. The mean was established as office BP. Brachial artery waveforms were then automatically recorded at 20-min intervals. Suitably sized cuffs were used according to the arm circumference measured in each study visit. All patients had recordings of good technical quality (≥70% valid readings). If not, a new ambulatory-BP-monitoring (ABPM) was repeated within 1 week.

##### Laboratory Analyses


*Urinary Albumin Excretion*


Urinary albumin excretion (measured by turbidimetry; lower detection limit: 0.3 mg/dl; intra-assay and inter-assay variation coefficients: 1.3% and 4.3%, respectively) was determined before BS and at the determined follow-up time-points and measured as the average of urinary albumin/creatinine ratio (ACR) from 2 fresh first-morning-void urine samples obtained on separate days. Microalbuminuria was defined as an ACR ≥ 30 mg/g. 


*Serum Creatinine and eGFR*


Serum creatinine (SCr) was measured by an enzymatic modified Jaffe reaction (CREA; Roche Diagnostics) using the Hitachi Modular System Analyzer (Roche Diagnostics), consistent with the current National Kidney Disease Education Program recommendations for standardizing SCr measurement [11]. The intra-assay coefficient of variation was 2.3%. 

There is no current agreement as to the best method to estimate the GFR in individuals with severe obesity. However, in a recently reported study [12], the modified Cockcroft-Gault (CG) equation performed best in both the overall population and the obese subgroup in terms of strength of correlation, mean bias and accuracy, as compared to both the IDMS (isotope dilution mass spectrometry) traceable simplified Modification of Diet in Renal Disease [13] and the Chronic Kidney Disease-Epidemiology Collaborative equations [14]. We obtained equivalent results to that report. Therefore, although we initially performed the analyzes by obtaining the eGFR using the three formulas separately, in the final analysis we only report data on eGFR according to the CG equation, with adjustments for body surface area.


*Renin-Angiotensin-Aldosterone System (RAAS) Components*


Plasma renin activity (PRA) and plasma aldosterone concentration, as well as angiotensin-converting enzyme (ACE) and angiotensin-converting enzyme-2 (ACE2) activities, were measured by validated laboratory methods [15]. Details on assay performance are reported in Appendix B.


*Adipokines and Inflammatory Parameters*


Leptin, adiponectin, and other cytokines and inflammatory markers, e.g., resistin, angiopoietin-2, MCP-1 and high-sensitivity C-reactive protein (hs-CRP), were also determined. See Appendix C.

##### Surgical Techniques

Either laparoscopic Roux-en-Y gastric bypass (LRYGB) or laparoscopic sleeve gastrectomy (LSG) were performed, and any of these two were chosen for each patient based on clinical criteria and the consensus of the Bariatric Surgery Unit. In this line, LSG was preferred in younger patients, in those with a BMI ranging from 35–40 kg/m^2^, as a first-step treatment in cases with a body mass index (BMI) > 50 kg/m^2^ and when drug malabsorption was to be avoided [16]. The LRYGB technique involved a 150-cm antecolic Roux limb with 25-mm circular pouch–jejunostomy and exclusion of 50 cm of the proximal jejunum. In LSG, the longitudinal resection of the stomach from the angle of His to approximately 5 cm proximal to the pylorus was performed using a 36-French bougie inserted along the lesser curvature.

#### 2.1.3. Statistical Analyses

Elementary statistical methods were applied with statistical package SPSS for Windows version 25.0 (Cary, NC, USA). The normality assumption for continuous variables was tested through the Kolmogorov–Smirnov test. Variables fulfilling this normality assumption were summarized as the mean ± S.D. or the median (interquartile range, IQR) otherwise. Categorical data were presented as frequencies and percentages. Comparisons of analyzed variables between two observed periods were carried out by paired t tests or Wilcoxon signed rank tests. Pearson or Spearman correlation coefficients were used for testing bivariate correlations as appropriate. Separate linear mixed-models were built for both the variation of eGFR-CG and the variation of albuminuria. All variables which deviate from the normal distribution were log-transformed (ln) before being introduced into the model. In these models, data were expressed as regression coefficient, 95% confidence interval (95% CI) and *p*-value. A change was considered significant if the two-side alpha level was ≤0.05. We used the statistical package SPSS for Windows version 25.0 (Cary, NC, USA), and STATA package version 15 (STATA Corp., College Station, TX, USA), for statistical analysis.

About one-third of the study population received treatment with one or more drugs that interfered with RAS. Given that these drugs could introduce a bias by interfering with renal function parameters, the main analyses were performed separately in both the entire cohort and in the untreated patients.

## 3. Results

Sixty-two patients completed the BARIHTA study, and information on renal changes was available for all of them at the baseline and follow-ups. A flowchart is supplied (Appendix A). Baseline characteristics are shown in Table 1. Fifty-five patients (89%) had hyperfiltration, i.e., eGFR > 120 mL/min/1.73 m^2^, before BS. None of the patients had chronic kidney disease based on the estimated glomerular filtration rate, nor did any of them die at follow-up.

### 3.1. Changes in Renal Function Parameters

The estimated GFR decreased from 155.9 ± 36.3 to 127.7 ± 27.4 mL/min/1.73 m^2^ (*p* < 0.001), reflecting the hyperfiltration that characterizes patients with severe obesity. In addition, the albuminuria, as measured by the log-transformed albumin-creatinine ratio (lnACR), decreased from 1.85 ± 1.08 to 1.56 ± 0.73 (*p* = 0.056). Both the estimated GFR (Figure 1A) and the albuminuria (Figure 1B) experienced a progressive decrease from baseline to the final observation 12 months after BS at all follow-up times (*p* for trend <0.001 for both). 

### 3.2. Changes in Body Weight, Body Mass Index, and Waist Circumference

Overall, there was a median (IQR) 30.5% (26.2–34.4) reduction in body weight 12 months after BS. Body mass index was 42.7 ± 5.6 kg/m^2^ at baseline and 29.7 ± 4.8 kg/m^2^ one year after BS (*p* < 0.001). Appendix A shows the mean (error bars 95% CI) body weight at each follow-up point. Waist circumference was 132.5 ± 12.0 cm and 105.5 ± 13.3 cm, at baseline and 12 months after BS, respectively (*p* < 0.001). A waist circumference (cm) decrease was confirmed in both men (140.2 ± 14.3 vs. 111.4 ± 16.3) and women (129.8 ± 10.1 vs. 103.4 ± 11.7), (*p* < 0.001 for both comparisons). 

### 3.3. Variation of Blood Pressure

As previously reported [10], office systolic and diastolic BP, both central and peripheral, significantly decreased at 12 months, even though more than 60% of the cohort were normotensives. In addition, central 24-h SBP decreased at 12 months, with a mean of (95% confidence interval) −3.1 mmHg (−5.5 to −0.7), *p* = 0.01 after adjustment for age and sex.

There was a statistically significant correlation between the variation at 12 months of peripheral 24-h systolic blood pressure (SBP) and the variation of both body weight (rho = 0.453, *p* = 0.001) and waist circumference (rho = 0.316, *p* = 0.030).

### 3.4. Changes in Glucose Metabolism Parameters and in RAS Components

Figure 2 shows the overall mean (95% CI) of fasting glucose before BS (0 months) and at 3, 6 and 12 months of follow-up. As noted, there is an initial decrease at 3 months that is maintained throughout the first year of follow-up (*p* for trend < 0.001).

Figure 3 shows the overall mean of glycosylated hemoglobin, fasting insulin and HOMA-IR (homeostasis model assessment-estimated insulin resistance index) before BS (0 months) and at 3, 6 and 12 months of follow-up. There is a statistically significant decrease of these three parameters throughout the first year of follow-up (*p* for trend < 0.001 for all).

The changes at 12 months after BS in various components of the RAS were also assessed. As expected, there was a statistically significant decrease in plasma renin activity and aldosterone plasma concentration, as well as an increase in the ACEactivity/ACE2 activity ratio after BS (see Table 2). These changes showed a tendency to persist when the subgroup of patients without a RAS blockade treatment that could interfere with the components of the RAS was analyzed separately.

There were statistically significant changes 12 months after BS in the explored adipokines and inflammatory markers (Appendix A). As seen, leptin and hs-CRP decreased, while adiponectine and angiopoietin-2 experienced an increase 12 months after BS.

### 3.5. Independent Correlates of Repeated Measurements of Renal Parameters

Separate mixed-models were built for both the variation of eGFR-CG and the variation of albuminuria (Table 3). The tested independent variables were those that were clinically relevant and that showed statistically significant changes at follow-up, mainly body weight, peripheral 24h-systolic BP, several components of the RAS, various adipokines and inflammatory markers and different parameters of the glucose metabolism. 

The models with a better performance show that the statistically significant independent variables for eGFR (Table 3A) were body weight and plasmatic aldosterone concentration, in both all patients and the subgroup of untreated patients. Regarding the variation of albuminuria (Table 3B), the main independent variable was the glycosylated hemoglobin. Similar results were found when the same models were tested including the HOMA insulin-resistance index instead of the glycosylated hemoglobin (see Appendix A). Otherwise, the variation of leptin, adiponectin, angioietin2 and hs-CRP lost statistical significance when included in the models.

## 4. Discussion

The main finding of this study is that in obese patients undergoing BS the mechanisms by which the estimated glomerular filtration rate and albuminuria return to normal values are possibly different. Thus, we demonstrate that normalization of eGFR is associated with a decrease in both body weight and plasma aldosterone concentration, while a decrease in albuminuria is directly correlated with an improvement in glucose metabolism.

Weight excess is associated with an altered renal haemodynamic profile, i.e., an increased GFR relative to effective renal plasma flow, resulting in an increased filtration fraction [17]. Hyperfiltration is the hallmark of obesity-associated renal dysfunction, leading to the onset of microalbuminuria, even before major structural changes occur [18]. Characterizing the renal function of patients with severe obesity and looking for the mechanisms underlying their evolution after BS are crucial challenges, especially to prevent obesity-related kidney damage. Controlled and noncontrolled studies have shown that BS decreases eGFR [4,7,19] and albuminuria [8,19,20], suggesting that BS alleviates hyperfiltration. 

Several studies have investigated the mechanisms likely to be responsible for renal changes observed in obese patients. A large number of these studies refer to hyperfiltration in general terms, focusing mainly on eGFR, especially with regard to the consequences of BS on renal function. Some of them found a relationship between the decrease in eGFR after BS and an improvement in the toxic adipokine profile observed in these patients [4,9]. In our cohort, we found a statistically significant decrease in leptin and in hs-CRP, as well as an increase in adiponectin, but these changes did not remain significant in the multivariate analyses after adjusting for other variables. On the contrary, we found that body weight and plasma aldosterone concentration decreases were the two factors that showed the strongest correlations with the restoration of eGFR to almost normal values. Some other authors have shown an association between the decrease in the percentage of high fat mass and the decrease in eGFR after BS. Of note, there is a general agreement that weight loss, but not the mechanism through with weight loss is achieved (i.e., type of surgery), is an independent predictor of restoration of renal function and prevention of chronic kidney disease [4,9,21,22]. Focusing on the role of aldosterone in the amelioration of hyperfiltration after BS in patients with obesity, it has been reported that activation of the RAS together with other mechanisms mediate increased renal sodium reabsorption in obesity-induced hypertension, high blood pressure being one of the possible causes of renal damage in these patients [23]. However, we must remark that our results are confirmed in the subgroup of untreated normotensive patients, therefore pointing to a hypertension-independent role of the RAS in renal damage in obese patients and its normalization after BS. It is suggested that the normalization of the eGFR after BS may be related to restoration in homeostasis of the RAS [24], and this is supported by experimental studies that show evidence of tubule-glomerular feedback resetting [25]. Thus, the correction of the dysregulated tubule-glomerular feedback may be the reason for those beneficial effects observed after BS.

The second relevant finding of this study is the close relationship between decreased albuminuria and improvement in all parameters of glucose metabolism. Indeed, most of the reported investigations regarding decreases in albuminuria after BS are based on cohorts of diabetic patients [20,26,27,28]. Podocyte dysfunction is considered of pivotal importance in the genesis of high albuminuria in both obesity and diabetes. Mechanical stress in the podocytes, secondary to glomerular hypertension, induces the differentiation and adhesion of the podocytes by reorganizing the actin cytoskeleton of the podocyte and compromising the size of the selective barrier of the slit diaphragm [8]. Here we show that improved glucose metabolism is an independent determinant of albuminuria regression, regardless of weight loss and blood pressure decrease and RAS changes, in a cohort where only 11% of patients were diabetic. Regarding the role of inflammation in decreasing albuminuria after BS, some authors have highlighted a possible role [9]. We also explored this mechanism, but although we found a correlation between changes in resistin and albuminuria, this cytokine no longer remained significant after adjusting for other variables.

We need to point out some limitations of the study. First, the GFR was estimated rather than measured directly. However, it should be noted that we determined the eGFR at various points over the 12 months and observed a significant decrease at any given time. In addition, although it has been suggested that weight loss-related changes in muscle mass influence eGFR to some extent, it is unlikely that a gradual decrease in eGFR over 12 months of follow-up will be accompanied by a continued decrease in muscle mass during this time. Second, for technical reasons, we were unable to determine interleukin-6 in most samples, so these data were excluded from the final analyses. Although it is a well-known marker of inflammation, we have complete data on hs-CRP, another hallmark of inflammation, and it was finally rejected that it played an important role in renal changes after adjustment for other variables. Finally, we cannot ignore the fact that the possible role of the RAS in changes in eGFR and glucose metabolism in changes in albuminuria are only hypotheses that should be confirmed in a randomized clinical trial.

## 5. Conclusions

In conclusion, our study demonstrates that weight loss and especially decreased aldosterone levels are the main factors associated to the restoration of the abnormally increased eGFR seen in obese patients after BS. This highlights the importance of the possible role of the RAS in renal hemodynamic alterations in obese patients that ultimately impair renal function. On the other hand, we show that increased albuminuria in obese patients is related to impaired glucose metabolism, regardless of blood pressure and weight loss.

## Figures and Tables

**Figure 1 jcm-11-00728-f001:**
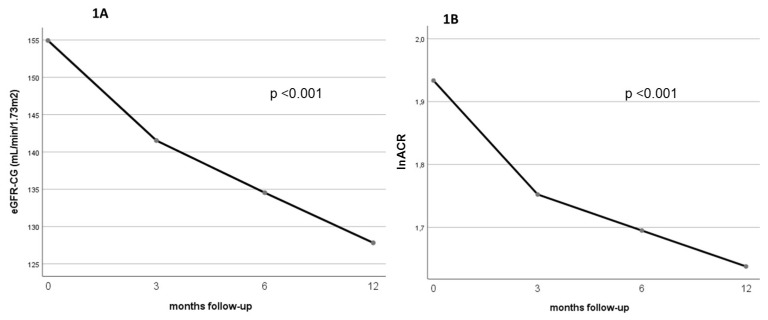
The changes in eGFR (**1A**) and albuminuria (**1B**) from baseline (before BS) to 12 months post-BS, with mid-points at 3 and 6 months. eGFR-CG = estimated glomerular filtration rate by the Cockcroft-Gault equation; lnACR = neperian logarithm of albumin-creatinine ratio. Values of both eGFR-CG and lnACR are given as mean ± SD.

**Figure 2 jcm-11-00728-f002:**
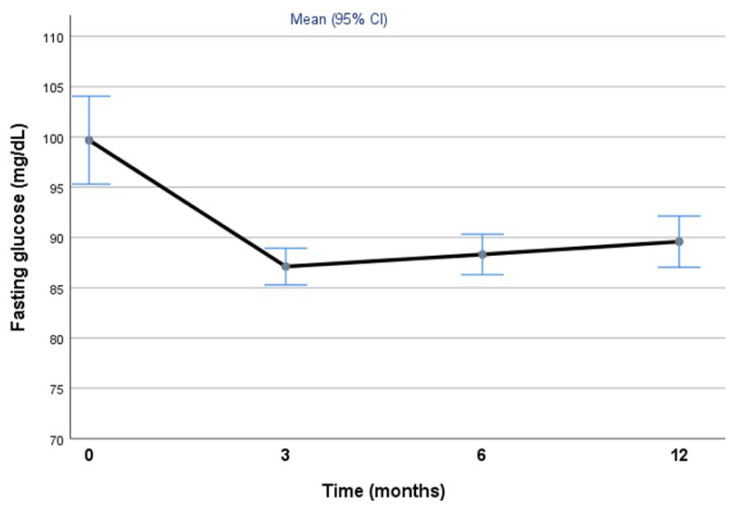
The changes in fasting glucose from baseline (before BS) to 12-months of follow-up, with mid-points at 3 and 6 months. Values of fasting glucose are given as the mean and corresponding 95% confidence intervals.

**Figure 3 jcm-11-00728-f003:**
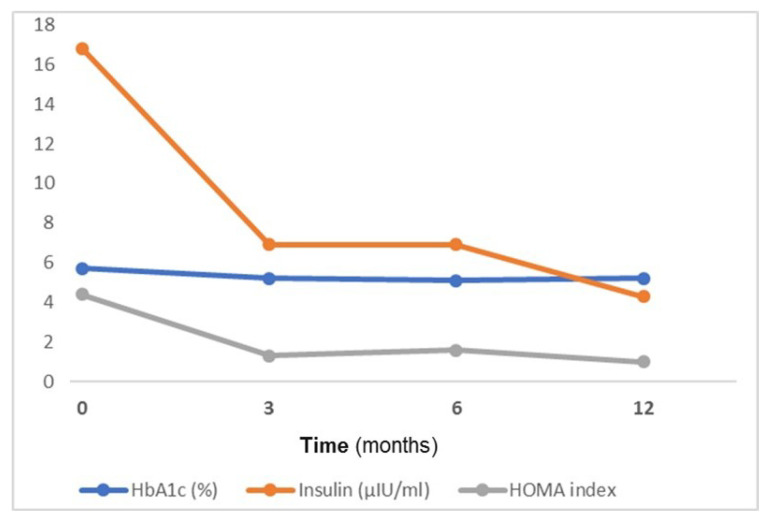
The changes in glycosylated hemoglobin, fasting insulin and HOMA-IR index from baseline (before BS) to 12-months of follow-up. HbA1c = glycosylated hemoglobin; HOMA = homeostasis model assessment-estimated insulin resistance.

**Table 1 jcm-11-00728-t001:** The baseline clinical characteristics.

Age, Year (Mean ± S.D.)	42.1 ± 9.3
Sex, women, *n* (%)	48 (77.4)
Body weight, kg (mean ± S.D.)	117.4 ± 18.9
Waist circumference, cm (mean ± S.D.)	132.3 ± 11.5
Body mass index, Kg/m^2^ (mean ± S.D.)	42.6 ± 5.5
Race, *n* (%) -Caucasian -African -Hispano-American	56 (90.3) 1 (1.6) 5 (8.1)
Current smokers, *n* (%)	17 (27.4)
Surgical procedure, *n* (%): -Sleeve gastrectomy -Roux-en-Y gastric bypass	27 (43.5) 35 (56.5)
Hypertension, *n* (%)	24 (38.7)
Type 2-Diabetes Mellitus, *n* (%)	7 (11.3)
Chronic kidney disease *, *n* (%)	0 (0)
Previous major vascular event, *n* (%)	3 (4.8)

* estimated glomerular filtration rate <60 mL/min/1.73 m^2^.

**Table 2 jcm-11-00728-t002:** The changes in the RAS components 12 months after bariatric surgery.

	All Patients (*n* = 62)			Patients without Antihypertensive Treatment (*n* = 42)		
	**Before BS**	**12-Months** **Post-BS**	** *p* **	**Before BS**	**12-Months** **Post-BS**	** *p* **
**PRA *,** **ng/mL/h**	0.8 (0.3; 1.3)	0.45 (0.2; 0.9)	**0.010**	0.85 (0.38; 1.3)	0.5 (0.2; 1.0)	0.074
**Aldosterone *, ng/dL**	87.8 (56.8; 154)	65.4 (56.8; 84.6)	**0.003**	81.6 (56.8; 110)	65.1 (56.3; 82.3)	0.090
**ACE activity, RFU/µL**	1244.1 ± 341.3	1287.3 ± 360.7	0.370	1272.3 ± 327.2	1295.6 ± 307.4	0.710
**ACE2 activity *, RFU/µL/h**	7.9 (5.8; 10.8)	6.9 (5.4; 10.8)	0.070	7.7 (5.8; 10.7)	6.9 (5.0; 11.3)	0.151
**ACE act./ACE2 act.**	164.5 ± 77.9	187.5 ± 78.4	**0.016**	172.7 ± 83.5	188.2 ± 74.1	0.131

(*) Data shown as median [interquartile range]. ACE = angiotensin converting enzyme; ACE2 = angiotensin converting enzyme 2; BS = bariatric surgery; PRA = plasma renin activity; RAS = renin-angiotensin system; RFU = relative fluorescence units.

**Table 3 jcm-11-00728-t003:** The determinants of the variation of eGFR (3A) and of the variation of albuminuria (3B).

**3A**						
**eGFR**	**All Patients (*n* = 62)**			**Patients without Antihypertensive Treatment** **(*n* = 42)**		
	**Coeff.**	**95% CI**	***p*-value**	**Coeff.**	**95% CI**	***p*-value**
**Months FU**	−0.42	−1.16, 0.32	0.267	−0.69	−1.59, 0.22	0.135
**Body weight, Kg**	0.71	0.46, 0.96	**<0.001**	0.62	0.32, 0.93	**<0.001**
**24h-systolic BP, mmHg**	0.26	−0.04, 0.56	0.089	−0.03	−0. 42, 0.36	0.887
**Aldosterone, ng/dL**	−0.11	−0.15, −0.07	**<0.001**	−0.07	−0.13, −0.02	**0.005**
**HbA1c,** **%**	1.24	−3.84, 6.31	0.633	2.98	−2.52, 8.48	0.288
**3B**						
**lnACR**	**All Patients (*n* = 62)**			**Patients without Antihypertensive Treatment** **(*n* = 42)**		
	**Coeff.**	**95% CI**	***p*-Value**	**Coeff.**	**95% CI**	***p*-Value**
**Months FU**	−0.01	−0.03, 0.02	0.539	0.00	−0.02, 0.03	0.788
**Body weight, Kg**	−0.00	−0.01, 0.01	0.641	0.00	−0.01, 0.01	0.609
**24h-systolic BP, mmHg**	0.02	0.01, 0.03	**<0.001**	−0.00	−0.02, 0.01	0.582
**Aldosterone, ng/dL**	0.00	−0.00, 0.00	0.107	0.00	−0.00, 0.00	0.608
**HbA1c,** **%**	0.27	0.09, 0.45	**0.004**	0.24	0.06, 0.42	**0.009**

BP = blood pressure; eGFR = estimated glomerular filtration rate; FU = follow-up; HbA1c = glycosylated hemoglobin; lnACR = log-transformed albumin-creatinine ratio.

## Data Availability

Not applicable.

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
