# Peer review of "Exploring Renal Changes after Bariatric Surgery in Patients with Severe Obesity"

_jcm, 2022, doi:10.3390/jcm11030728_

Round 1

Reviewer 1 Report

In this study, Dr. Oliveras and colleagues aimed to explore the mechanisms of obesity-associated renal changes after bariatric surgery. Overall, this is a nicely written study. Nonetheless, I have some comments and suggestions:

  • In this study, the authors "only" performed a prospective observational study in 3, 6, 12 months post-BS. Unfortunately, this type of study is not adequate to explain the mechanisms of eGFR changes as intended because to do so, the authors need to perform an RCT and even an experimental study. Here, the authors could show association / correlation but I am afraid nothing more than that. So, please refrain from claiming anything about mechanisms and please revise the entire manuscript. 
  • I think the title and aim has to be better clarified. I was confused whether this study was about the renal changes in "obesity that happened post-BS" or the renal changes post-BS in general. 
  • Line 27 and the rest: Please always use mean+/- SD or median +/- IQR. Otherwise, please justify the reason for using mean +/- IQR.
  • "Both estimated GFR (eGFR) and albuminuria decreased from baseline at all follow-up times (p-for-trend <0.001 for both)." Please add the values, especially for eGFR because this sentence seems to indicate that the renal function is worsened following BS (due to a reduction of eGFR), which is contradictory to what they showed previously in a meta-analysis by Li et al, REF[6]. By adding the values, readers would understand that BS corrects the hyperfiltration-related pathological increase of GFR.
  • Have the authors performed an adjustment to body surface area when calculating the eGFR?
  • The introduction of BS is lacking. Please add a paragraph discussing what is BS and the benefits in reducing obesity-related morbidity and mortality, including MACE. The authors could benefit from this meta-analysis (PMID: 34684569). 
  • I think the authors need to add the list of medications in Table 1 since the use of drugs could also affect the findings. Also, for the race, please specify the rest (other than caucasians). 
  • "we found that body weight and plasma aldosterone concentration decreases were the two factors that determined the restoration of eGFR to almost normal values." which findings supported this statement? As far as I understand, the authors found a correlation but it could not tell about the causal effect. Please clarify this statement. 

In general, I strongly suggest the authors to be careful with the terminology "mechanisms". Association / correlation is not enough to define a dynamic change / process because it needs a situation that can be controlled to understand the mechanism of something. The fact that they might be a part of the mechanisms cannot be excluded but still it is deemed inappropriate to unravel a pathophysiology from an observational study. So, please revise everything.

Reviewer 2 Report

General comments

I have read the article entitled “Deciphering Mechanisms for Renal Changes Related to Obesity After Bariatric Surgery.” By Oliveras et al. The aim of the study was to seek for potential mechanisms of obesity-related renal changes post-BS. Sixty-two patients with severe obesity were prospectively examined before and 3, 6 and 12 months post-BS. Estimated GFR (eGFR) and albuminuria decreased from baseline at all follow-up time. There was a mean 30.5% reduction in body weight. Plasma glucose, glycosylated hemoglobin, fasting insulin and HOMA-index decreased at 3, 6 and 12 months of follow-up. Plasma aldosterone concentration also decreased at 12-months. Both leptin and hs-CRP decreased and adiponectin levels increased at 12-months post-BS. Linear mixed-models showed that body weight and plasma aldosterone were the independent variables for changes in eGFR. Conversely, glycosylated hemoglobin was the only independent variable for changes in albuminuria. The authors conclude that body weight and aldosterone are the main factors that mediate eGFR changes in obesity and BS, while albuminuria is associated with glucose homeostasis.

I think that this is an interesting mechanistic study about an important clinical problem, such as the mechanisms for renal changes related to obesity after bariatric surgery. The article was well designed and addressed in a balanced way.

Specific comments

I have several specific comments, all of which could be addressed with some minor revision to the manuscript.

Minor

The title and the aim of the article are not exactly the same. Pages 1 and 2.

Please detail clearly the indications for BS. Page 2 line 70.

I think you should probably include median with IQR, instead of mean? Page 1 line 27 and Page 5 line 191.

The figure captions should be more clear, and include if it is the mean or the median. The figures itself should include SD or IQR. Pages 5 and 6.

In my view, the main limitation of the present study is that, as you point out, the GFR was estimated rather than measured directly and body weight is included in the CG equation. In addition, adiposity should be more correctly presented as BMI instead of body weight and the correlation studies should be between BMI and eGFR. Could you comment on this aspect? Pages 7 and Discussion.  

There are some typographical and grammatical errors throughout the text.

Round 2

Reviewer 1 Report

Thank you for the responsive answers. I have no further comment except for this very small suggestion. 

  • "Recently, obesity-induced hyperfiltration and albuminuria have been shown to be reversible after bariatric surgery (BS)" The authors could add "recently".

Author Response

Thanks again to the reviewer for his/her appreciation.

According to the comment, we have now added "Recently, ..." in the suggested point in page 2, now in yellow.

Best regards,